# Gamma Irradiation Processing on 3D PCL Devices—A Preliminary Biocompatibility Assessment

**DOI:** 10.3390/ijms232415916

**Published:** 2022-12-14

**Authors:** Fernando Guedes, Mariana V. Branquinho, Sara Biscaia, Rui D. Alvites, Ana C. Sousa, Bruna Lopes, Patrícia Sousa, Alexandra Rêma, Irina Amorim, Fátima Faria, Tatiana M. Patrício, Nuno Alves, António Bugalho, Ana C. Maurício

**Affiliations:** 1Veterinary Clinics Department, Abel Salazar Biomedical Sciences Institute (ICBAS), University of Porto (UP), Rua de Jorge Viterbo Ferreira, n° 228, 4050-313 Porto, Portugal; 2Animal Science Studies Centre (CECA), Agroenvironment, Technologies and Sciences Institute (ICETA), University of Porto, Rua D. Manuel II, Apartado 55142, 4051-401 Porto, Portugal; 3Associate Laboratory for Animal and Veterinary Science (AL4AnimalS), 1300-477 Lisbon, Portugal; 4Centre for Rapid and Sustainable Product Development (CDRSP), Polytechnic Institute of Leiria, 2411-901 Leiria, Portugal; 5Department of Pathology and Molecular Immunology, Abel Salazar Institute of Biomedical Sciences (ICBAS), University of Porto (UP), Rua Jorge Viterbo Ferreira, n° 228, 4050-313 Porto, Portugal; 6Institute of Research and Innovation in Health (i3S), University of Porto (UP), Rua Alfredo Allen, 4200-135 Porto, Portugal; 7CUF Tejo Hospital and CUF Descobertas Hospital, 1350-070 Lisbon, Portugal; 8Center of Studies for Cronic Diseases (CEDOC), NOVA Medical School, 1150-082 Lisbon, Portugal

**Keywords:** gamma irradiation, 3D printing, polycaprolactone, mesenchymal stem cells, cytocompatibility, biocompatibility

## Abstract

Additive manufacturing or 3D printing applying polycaprolactone (PCL)-based medical devices represents an important branch of tissue engineering, where the sterilization method is a key process for further safe application in vitro and in vivo. In this study, the authors intend to access the most suitable gamma radiation conditions to sterilize PCL-based scaffolds in a preliminary biocompatibility assessment, envisioning future studies for airway obstruction conditions. Three radiation levels were considered, 25 kGy, 35 kGy and 45 kGy, and evaluated as regards their cyto- and biocompatibility. All three groups presented biocompatible properties, indicating an adequate sterility condition. As for the cytocompatibility analysis, devices sterilized with 35 kGy and 45 kGy showed better results, with the 45 kGy showing overall improved outcomes. This study allowed the selection of the most suitable sterilization condition for PCL-based scaffolds, aiming at immediate future assays, by applying 3D-customized printing techniques to specific airway obstruction lesions of the trachea.

## 1. Introduction

Several fabrication techniques have been used to create clinically applicable scaffolds with uniform pore size and control over geometry. Additive manufacturing (AM), also known as 3D printing, enables customized fabrication of 3D constructs based on computer-aided design software or images obtained from computer tomography and magnetic resonance. There are several AM techniques, among which only some techniques are widely applied in the medical industry [1]. The most widely used are droplet-based printing [2], and extrusion-based printing [3]. Tissue engineering (TE) with 3D printing is focused on two different perspectives: functional biomaterials for tissue implantation and tissue models for disease [1]. This study will focus on biomaterials based on the first option.

Scaffolds have found their place as templates for cell interaction, providing physical support to the fresh developed tissue [3]. Furthermore, scaffolds can function as delivery vehicles to incorporate essential growth factors and biomolecules to control and enhance tissue growth [4]. The aim of 3D bioprinting is to mimic the natural cellular architecture by depositing materials and cells to restore the normal structure and functionality of complex tissues. TE scaffolds are fabricated in two major methods: printing with cells mixed in ink or gel, or seeding cells onto scaffolds post printing [4].

The role of airway TE, a field of regenerative medicine, is to develop biological substitutes that can restore, maintain, or improve tissue functions. As a “simple” cylinder and with a relatively main function, which is to conduct the air, the trachea was initially considered as a good starter organ for TE, and historically many attempts have been made with autografts, allografts, and prosthetic materials [5,6]. The ideal tissue-engineered scaffold for airway would be capable of promoting exogenous cell engraftment and endogenous cell ingrowth, proliferation, and appropriate differentiation, while maintaining a patent airway.

In the context of airway TE, central airway obstruction (CAO) represents a pathological condition that leads to airflow limitation of the trachea, main stem *bronchi*, *bronchi intermedius* or *lobar bronchus*. That represents an important clinical impairment and can be caused either from benign or malignant diseases [7,8]. The incidence is still not well-known, but it tends to be underdiagnosed [9]. While the surgical procedure is considered the gold standard, several clinical conditions concerning the clinical condition of the patient, the degree and the type of CAO could compromise the feasibility of the standard therapy [10]. In those cases, interventional bronchology with its accessory techniques, such as laser, cryotherapy dilatation or airway stenting, could represent a good option [11,12]. Airway stenting could be used in both benign and malign diseases, but placing the stent must be a very well-balanced decision because, while on the one hand, the re-occlusion is prevented, on the other hand, the procedure presents several disadvantages, such as stent migration, mucous plugging, recurrent infection or fistulation [10,13]. There are two main types of airway stenting concerning the material: silicone stents and metallic stents. New custom-made and bioabsorbable airway stents made of different biomaterials are under investigation and have been placed in humans. Polydioxanone is the one that has been most often used and for the longest period [14]. Other biomaterials such as polylactic acid, polyglycolic acid, polycaprolactone, polyurethane or polyamide are under investigation [3,15]. Biodegradable polymers, such as polycaprolactone (PCL), are increasingly used for 3D printing of scaffolds. This material offers great advantages such as biocompatibility, biodegradability, and good mechanical properties [16,17]. Polycaprolactone (PCL) is a hydrophobic, biodegradable polyester with a molecular weight usually between 3000 and 80,000 g/mol, a density of 1.146 g/mL at 25 °C, a low melting point of around 60 °C and a glass transition temperature of about −60 °C. It is a semi-crystalline polymer, the crystallinity of which tends to decrease with increasing molecular weight [18,19,20]. A study by Castilla-Cortázar et al. calculated a percentage crystallinity in pure PCL of 39.1 using differential scanning calorimetry analysis [21]. The total degradation of PCL is considerably affected by the molecular weight and crystallinity of the material and can vary between one and four years. It can be used as a polymeric plasticizer because of its ability to lower elastic modulus and soften other polymers. Its surface is chemically suitable for cell attachment, proliferation, and differentiation, and its degradation by-products are nontoxic and are usually metabolized and eliminated via natural pathways.

Woodward and his group studied the in vivo and intracellular degradation of PCL and reported that degradation first occurred with nonenzymatic bulk hydrolysis, and a transient initial inflammatory response occurred only for the first 2 weeks. After 9 months, when the molecular weight had significantly reduced, a loss in mass emerged and PCL did fragment [18]. The most important properties of a bioabsorbable scaffold are the degradation rate, mechanical strength, and ability to support cell growth. Gamma rays at 30.8 kGy significantly decrease the rate of degradation of PCL without affecting molecular weight or cells attachment and growth. Other studies of various doses of gamma irradiation and the impact on PCL revealed a shift toward a lower molecular weight in a dose-dependent manner accompanied by an increase in both the melting point and crystallinity [18]. In addition, D’alelio et al. report typical critical doses (200–300 kGy (20–30 Mrad)) required to initiate gel formation in linear polyesters containing 3–7 methylene groups per ester group [22]. Narkis et al. report g.p.c molecular weight distributions as a function of the irradiation dose in the pre-gel stage of PCL (Dose < 260 kGy (26 Mrad)). Number and weight average molecular with CH2/COO ratio of infinity undergoes only crosslinking reactions, and gel formation is found from the very beginning of the irradiation process if the starting molecular weight is not too low. Typical critical dose of 260 kGy (26 Mrad) is required to initiate gel formation in PCL (CH2/COO = 5/1) [23]. In spite of gamma rays having induced chain scission and crosslinking, these works are focused mainly on much higher doses than those required for sterilisation.

However, to be approved for human implantation, they must be in sterile conditions. Numerous techniques have been researched, all of them subject to limitations [24]. The most frequently used are ethylene oxide, beta radiation, gamma radiation, peracetic acid and hydrogen-peroxide plasma. Steam and heat sterilization are not feasible in this polymer as it has a melting point of 59–64 °C. Ethylene oxide could be an option, but it is known to soften PCL, and its residual vapors left in the device found to be mutagenic and carcinogenic [25]. Thus, ionizing is likely the method of choice and gamma radiation represents the most extensively studied sterilization method for PCL [26,27]. Gamma radiation is highly penetrative and kills bacteria by breaking down bacterial DNA, thereby inhibiting bacterial division. On the other hand, such photon-induced damage at the molecular level can also cause changes in the physical and chemical properties of the polymer. A minimum dose of 25 kGy is routinely applied for sterilization of many medical devices and biological tissues. As recommended by the International Organization of Standardization (ISO), the sterilization dose must be set for each type of product depending on its characteristics and the load of microbes [26].

The most important properties of a bioabsorbable scaffold are the degradation rate, mechanical strength, and ability to support cell growth. Gamma rays at 30.8 kGy significantly decrease the rate of degradation of PCL, without affecting molecular weight or cells attachment and growth. Considering mechanical properties, yield stress increased significantly but the stress at break did not. Scaffolds represent an important support rule in the airway, and as such are of the upmost relevance [23]. Augustine et al. report that a low radiation dose first would lead to improved PCL mechanical properties; however, higher doses would decrease them. Thus, results for the effect of gamma radiation on the mechanical properties are ambiguous and a general trend has not yet been established.

In this preliminary work the authors intended to study the cytocompatibility and biocompatibility of 3D-printed polycaprolactone (PCL)-based devices [28] after different gamma radiation conditions. The choice for the biomaterial relied on the premise that, in future assays, the device to be studied to promote regeneration of tracheal tissue must have a design that mimics the anatomical shape of trachea, must have mechanical strength and flexibility similar to the native trachea’s and porosity that allows good vascularization and cell proliferation. In addition, it must be biocompatible, biodegradable and non-immunogenic [29]. Furthermore, PCL allows the production of 3D devices with interconnected porous network and high reproducibility [30]. Following the biomaterial production and characterization studies, in vitro and in vivo assays were conducted to assess the gamma radiation effect on the device’s properties, the first applying mesenchymal stem cells (MSCs) from dental pulp tissue of human origin and the latter considering subcutaneous implantation on a rat animal model.

## 2. Results

### 2.1. Devices Characterization

Fourier-transform infrared spectroscopy (FTIR) spectra of the scaffolds sterilized with different conditions are depicted in Figure 1. All samples present similar results in terms of chemical structure, with the characteristic bands of pure PCL. The bands at 2865 and 2941 cm^−1^ are related to the symmetric and asymmetric stretching of the CH2 group. A strong absorbance at 1720 cm^−1^ also corresponds to a structural group of PCL, more precisely C=O stretching vibration of the ester linkages. Figure 2 shows the results of the SEM examination of the PCL scaffolds gamma irradiated up to 45 kGy. All samples present micropores in the filament surface, indicating that sterilization at 25, 35 and 45 kGy has no effect on the morphology of the filaments.

### 2.2. In Vitro Cytocompatibility Assessment

A PrestoBlue^TM^ cytocompatibilty assessment was conducted on the produced scaffolds, following sterilization. Scaffolds were divided into three groups, considering 3 different levels of sterilization: 25 kGy, 35 kGy and 45 kGy. A control group of the cell population was considered, by seeding cells directly to the well with no scaffold, so as to access cell normal behaviour and proliferation in culture. For each time-point, corrected absorbance values were obtained for each group and are presented in Figure 3 (upper panel) and Table 1.

Human dental pulp stem/stromal cells (hDPSCs) were employed in this assay, following previous works [30,31,32]. These cells’ population regenerative potential towards the osteogenic lineage has been established by Campos et al. [33,34] and was selected for the purpose of this study, as the authors intend to further analyse the regenerative potential of PCL-based devices for tracheal airway-obstruction cases.

Following the Presto Blue^TM^ cytocompatibility assessment, samples were processed for Scanning Electronic Microscopy (SEM). Seeded and unseeded devices were considered. Images are presented in Figure 4.

SEM analysis allowed the visualization of adhered cells to all the devices, which presented a fibroblast-like shape, normal morphology and adequate adhesion. Cells presented elongations of the cytoplasm, creating adhesion points between the device’s fibers, thus creating a 3D cellular network. No differences could be qualitatively established between the groups.

### 2.3. In Vivo Biocompatibility Assessment

Following the in vitro assessment, devices were further assessed in vivo for their biocompatibility, according to ISO 10993-6:2016 guidelines for Biological evaluation of medical devices, Part 6: Tests for local effects after implantation. Scaffolds were implanted subcutaneously on the dorsum of Sasco Sprague-Dawley rats and analysed after 7 and 15 days post-implantation time. Furthemore, samples were embedded in paraffin and analysed using SEM. The obtained images are presented in Figure 5. A semi-quantitative scoring analysis was performed, according to annex E of the referred guideline. Evaluation of the biological reaction to the devices included quantification of fibrosis, extent, inflammatory cells, necrosis, neovascularization, fat tissue infiltration, among others. Results are presented in Figure 6 and Figure 7 and Table 2. 

The macroscopical evaluation of the samples revealed no signs of hemorrage, infection or inflammation. All samples revealed, microscopically, minimal fibrosis extent, and non-detectable or rare necrosis or giant cells presence. Neoscularization was detected for all groups, as well as polymorphonucleated (PMN) cells, the latter decreasing in the latest timepoint. Following the pre-established criteria, all groups were considered biocompatible, presenting a score value contained in the “minimal or no reaction” category.

## 3. Discussion

FTIR analysis was performed before and after the use of different gamma irradiation conditions to evaluate any alterations in functional groups during sterilization. Comparing the spectrum of PCL scaffolds without sterilization with the spectra of sterilized PCL scaffolds, it can be seen that after gamma irradiation there were no evident modifications in the bands. These results corroborate those of Tapia-Guerrero et al. and Paula et al.’s works, which reported no significant changes in the PCL functional groups after sterilization using gamma irradiation [35]. SEM analysis was employed to determine the morphological features of the scaffolds. The microporosity observed on the surface of the filaments was caused by the material preparation method, more precisely due to the solvent addition [30]. Thus, gamma irradiation had no influence on the filament morphology of PCL scaffolds.

A thorough cytocompatibility analysis was performed, including a qualitative SEM analysis of the in vitro-seeded PCL-based scaffolds with hDPSCs. This assay allowed the visualization of cellular adhesion, as well as a cellular layer formation in the 3D scaffolds, with a uniform distribution (Figure 4), thus confirming that the analyzed scaffolds present potential cellular matrix conditions for 3D culture. Furthermore, SEM images of the subcutaneously implanted 3D PCL-based scaffolds in the rat animal model allowed visualization of the 3D structure integrity of the scaffolds (Figure 5), as well as tissue integration, which was validated by the histological scanning of the samples with H&E staining (Figure 6).

As for the quantitative analysis, a Presto Blue^TM^ viability assay confirmed the scaffolds’ cytocompatibility, thus sustaining the qualitative assessment by SEM analysis. All groups presented overall promising outcomes, with the 35 and 45 kGy presenting slightly better results, compared to the 25 kGy group. These results have already been identified in previous studies with the same tendency, that is, materials subjected to higher radiation levels showing a higher rate of cell proliferation and cytocompatibility [36,37]. These results confirm that gamma radiation has no adverse effects on cell proliferation and promotes cell proliferation in a dose-dependent manner. Although the reason for this observation is unclear, previous work seems to indicate that an increase in radiation dose is associated with changes in the hydrophilicity of materials [25]. Higher doses of radiation seem to promote a decrease in the contact angle, which in turn stimulates adhesion and consequent cell proliferation. However, a more extensive characterization of the devices would be necessary to confirm this biophysical change and fully justify the results. The control group was only considered to assess the cellular population’s health, proliferation and normal behavior in culture, and direct comparisons between this group should be taken carefully, as, in contrast with the tested groups, this control group is a 2D culture condition, since no matrix supporting biomaterial is considered. When considering tissue regeneration, 3D culture conditions are more reliable as regards mimicking the in vivo environmental conditions, when compared to 2D cultures. The latter are often associated with higher proliferation rates, but also with loss of diverse phenotype and metabolism alterations, thus compromising comparability with in vivo conditions [25,27]. For this matter, 3D culture conditions are more likely to mimic the in vivo settings, as cell–extracellular, and cell-to-cell interactions are more precisely evaluated. Further, cells have variable access to nutrients and oxygen, as in in vivo conditions.

Considering the biocompatibility assay, the rat animal model was employed, and the scaffolds were implanted subcutaneously on the animal dorsum, considering 7 and 15 days’ recovery period. A semi-quantitative evaluation was performed, according to ISO 10993-6: 2016 guidelines for “Biological evaluation of medical devices, Part 6: Tests for local effects after implantation”. A scoring system was applied (according to Annex E) when evaluating the biological response, as regards the fibrosis’ extent, changes in tissue morphology, necrosis presence, vascularization, fatty infiltration, and the presence of inflammatory cells. Upon euthanasia, ex vivo tissue presented no abnormalities, with no visible hemorrhage or inflammation/infection. A global histological score was calculated for each group and a score was determined for each experimental sample by subtracting the sham group effect, associated with the intrinsic healing capacity (Table 2 and Figure 7). Microscopically, minimal fibrosis and neovascularization were detected in all samples at 7 days post-implantation. Necrosis events and giant cells were rarely signaled, and mononuclear inflammatory cells (lymphocytes and macrophages) existed at a greater extent, when compared to polymorphonuclear cells, for all groups and timepoints. A slight increase in neovascularization and decrease in fibrosis was detected at 15 days post-implantation. According to the ISO 10993-6 scoring system, all samples were classified as “minimal to no reaction” at both timepoints, thus confirming its biocompatibility ability and suitability for in vivo implantation. In addition, several organs were further analyzed through a necropsy exam and microscopic examination to access the systemic effects. Different organs were considered, such as lungs, liver, heart, spleen, and pancreas. No alterations were detected.

## 4. Materials and Methods

### 4.1. Material Preparation and Devices Production & Characterization

Solid pellets of PCL (ø ~3 mm, MW 50,000, from Perstop Caprolactones (Cheshire, UK)) were dissolved in N,N Dimethylformamide (DMF, from Merck KGaA^®^, Darmstadt, Germany) with solvent-casting technique, using the amount of 1 g of PCL in 4 mL of DMF [29,38]. After the full dissolution of the PCL, the solution was deposited into petri dishes and left to dry in fume hood until solvent evaporation, for further use in a 3D printing system. In the current research, a biomanufacturing system (ANI Biomate Project) developed by CDRSP-IPLeiria was used. This system incorporates several techniques, such as micro-extrusion, multi-head dispensing and electrospinning [29,38]. The extrusion technique used in this study consisted of heating the material (PCL), and pneumatic and mechanical extrusion, i.e., compressed air and a rotating spindle were used for controlled deposition of the composite, respectively. The 3D cylindrical scaffolds of 10 mm diameter and 2.5 height were produced using the following design parameters: 0°/90° laying pattern, 0.35 mm pore size and 0.3 mm filament diameter. Regarding processing parameters, the material was heated to 80 °C and deposited using a screw rotation velocity of 15 rpm at a working speed of 300 mm/min.

The 3D polycaprolactone (PCL) scaffolds were sterilized with gamma radiation (25 kGy, 35 kGy and 45 kGy), in a Red Perspex Dosimeters. Fourier Transform Infrared Spectroscopy (FTIR) was performed in the sterilized PCL scaffolds using Alpha FT-IR spectrometer (Bruker, Kontich, Belgium) and Opus Software. Samples were analyzed at room temperature, in a spectral range of 400–4000 cm^−1^, with a resolution of 4 cm^−1^ in a total of 64 scans.

PCL scaffolds’ morphology was also observed using a scanning electron microscope (SEM) (VEGA 3, TESCAN, Kohoutovice, Czech Republic) that was operated at a voltage of 15 Kv. Before observation, the scaffolds were coated with gold–palladium.

### 4.2. Sample Sterilization

The 3D PCL scaffolds were sterilized with gamma radiation (25 kGy, 35 kGy and 45 kGy), in a Red Perspex Dosimeters. After sterilization, the surface morphology of all 3D constructs was analyzed using a scanning electron microscope (SEM) (VEGA 3, TESCAN) that operated at a voltage of 15 kV, after coating the powders with gold–palladium.

### 4.3. In Vitro Assays

#### 4.3.1. Cell Culture and Maintenance

Human dental pulp stem/stromal cells (hDPSCs) obtained from AllCells, LLC (Cat. DP0037F, Lot N° DPSC090411-01) were maintained in MEM α, GlutaMAX™ Supplement, no nucleosides (Gibco, 32561029), supplemented with 10% (*v*/*v*) fetal bovine serum (FBS) (Gibco, A3160802), 100 IU/mL penicillin, 0,1 mg/mL streptomycin (Gibco, 15140122), 2.05 µm/mL amphotericin B (Gibco, 15290026) and 10 mM HEPES buffer solution (Gibco, 15630122). All cells are maintained at 37 °C, 80% humidified atmosphere and 5% CO_2_ environment. Campos et al. previously described the characterization of these cellular populations [34].

#### 4.3.2. Cytocompatibility Evaluation

The cytocompatibility between the cellular system and the scaffolds was assessed with a Presto Blue^TM^ assay to determine the impact of the sterilization intensity on the adhesion and cellular viability. The Presto Blue^TM^ assay, a viability assessment, analyzes the permeability of cells to a resazurin-based solution. This solution allows the quantification of cellular viability, by modifying the media color after metabolization of the reagent by viable cells.

This assay was conducted as described in previous works [31,32,33]. Briefly, scaffolds are pre-hydrated in complete FBS medium for 24 h and then seeded through dynamic seeding, where the cellular suspension is incubated with the scaffolds in a roller bank, each with a 2.5 × 10^5^ cellular suspension, for 8 h, at 37 °C, 80% humidified atmosphere and 5% CO_2_ environment. Later, seeded scaffolds are transferred to a non-adherent 24-well plate and concealed with complete medium. Culture media is removed and fresh media are added to every cultured well, at each time-point (24, 72, 120 and 168 h). 10% (*v*/*v*) 10× Presto Blue^TM^ cell viability reagent (Invitrogen, A13262) is added to each well, and plates are incubated for 1 h. Following, 100 µL of media is transferred to a 96-well plate and absorbance is read at 570 and 595 nm. Dulbecco’s phosphate-buffered saline solution (DPBS, Gibco, 14190169) is used to wash and remove the reagent from the wells, prior to adding fresh media. Absorbances were read at 570 nm and 595 nm with a Multiskan^TM^ FC Microplate Photometer (Thermo Scientific^TM^, 51119000), following manufacturing instructions. For this assay, blank wells were considered, containing the respective scaffold, but no cells. Absorbance data were collected for each well by subtracting values obtained at 595 nm from values obtained at 570 nm. Data were further corrected by subtracting the average of the blank wells (average of 570–595 nm) from the absorbance values (570–595 nm) of each experimental well (seeded scaffolds).

A control of the cellular population was considered, where cells were seeded in 10% FBS supplemented media, directly in a tissue-treated 24-well plate, with a density of 7000 cells per cm^2^, so as to control cell normal growth and proliferation.

#### 4.3.3. Scanning Electronic Microscopy (SEM)

Further, seeded scaffolds were fixated for SEM analysis, as described in previous works [31,33]. Scaffolds were rinsed three times with a 0.1M HEPES (Merck^®^, PHG0001) buffer solution and left overnight in a fixative solution containing 2% glutaraldehyde (Merck^®^, G5882). A dehydration crescent alcohol series (50%, 70%, 95% and 99%) was conducted previously to the incorporation of hexamethyldisilazane (HMDS, Alfa Aesar, A15139). Samples were left overnight to evaporate remaining residues of the reagents.

Following, samples were coated with Au/Pd using sputtering (SPI Module Sputter Coater) for SEM analysis with a high resolution (Schottky) environmental scanning electron microscope with x-ray microanalysis and electron backscattered diffraction analysis, Quanta 400 FEG ESEM/EDAX Genesis X4M, in high vacuum mode.

### 4.4. In Vivo Biocompatibility Assessment

Animal testing assays were conducted in conformity with the Directive 2010/63/EU of the European Parliament and the Portuguese DL 113/2013 with previous approval from the ICBAS-UP Animal Welfare Organism of the Ethics Committee (ORBEA) and from the Veterinary Authorities of Portugal (DGAV). Humane endpoints in agreement with the OECD Guidelines (2000) were followed. The in vivo biocompatibility assessment was performed in adult male Sasco Sprague-Dawley rats (Charles River, Barcelona, Spain) weighing 250–300 g, as described in previous works [26,27,28]. An adequate environment for the animals was considered, with controlled temperature, humidity, and 12–12 h light/dark cycles. Feeding included standard chow and water *ad libitum*. For the surgical procedure, anesthesia was administered intraperitoneally: Xylazine/Ketamine (Rompun^®^/Imalgène 1000^®^; 1,25 mg/9 mg per 100 g b.w.), following an aseptic skin preparation of the dorsum. Incisions measuring 15–20 mm long were performed, and scaffolds were implanted subcutaneously, following incision suture. Animals were recovered, evaluated, and returned to their housing groups. Shams were considered, where the surgical access was performed but no medical device implanted. At 7 and 15 days’ recovery time, animals were subjected to anesthesia, as described above, and euthanized using lethal intra-cardiac injection (Eutasil^®^ 200 mg/mL, 200 mg/kg b.w.). Skin and subcutaneous tissue were collected and fixated in 4% formaldehyde (Merck^®^, 100496).

Following, samples were processed for histopathological analysis, and stained with hematoxylin-Eosin (H&E). Stained sections were analyzed with a Nikon microscope (Nikon Eclipse E600) equipped with ×2, ×4, ×10 and ×40 objectives and coupled with a photo camera (Nikon Digital Sight DS-5M) equipped with a lens (Nikon PLAN UW 2X/0.06). Evaluation followed ISO-10993-6:2016 guidelines, annex E, and included inflammatory infiltration, fibrosis, angiogenesis and/or necrosis surrounding the implant. A scoring system was established, following a semi-quantitative classification of the implants as “minimal or no reaction” (score 0,0 up to 2,9), “slight reaction” (score 3,0 up to 8,9), “moderate reaction” (score 9,0 up to 15,0) or “severe reaction” (score > 15).

### 4.5. Statistical Analysis

Statistical analysis was performed using GraphPad Prism version 6.00 for Mac OS x, GraphPad Software, La Jolla, CA, USA. Triplicates were considered and results are presented as mean ± standard error of the mean (SE). Analysis was conducted using two-way ANOVA analysis with Tukey multicomparison test. Differences were considered statistically significant at *p* ≤ 0,05. Results’ significances are presented through the symbol (*), according to the *p*-value, with one, two, three or four symbols, corresponding to 0.01 < *p* ≤ 0.05; 0.001 < *p* ≤ 0.01; 0.0001 < *p* ≤ 0.001; and *p* ≤ 0.0001, respectively.

## 5. Conclusions

Tissue engineering of customized 3D-printed medical devices relies on the premise of producing safe and implantable biomaterials, where the sterilization process plays an important role. Gamma irradiation is one of the most employed sterilization processes, although the most suitable radiation level must be previously established for each study, depending on various conditions, such as the biomaterials’ properties and medical condition in scope. The authors intend to investigate the potential of developing customized 3D-printed PCL-based scaffolds for tracheal occlusion in CAO scenarios. For this purpose, this preliminary study was conducted to assess the most suitable gamma radiation condition for PCL-based scaffolds sterilization. Higher levels, from 35 to 45 kGy, have presented better cytocompatibility outcomes, although 25 kGy presented equally good outcomes regarding biocompatibility after subcutaneous implantation in a rat animal model. Results suggest radiation levels of 35 kGy or 45 kGy to be safer and more suitable for the sterilization of these devices.

Considering mechanical properties, yield stress increased significantly but the stress at break did not. The scaffold represents an important support rule in the airway and this is of the upmost relevance [28]. Augustine et al. report a low radiation dose first would lead to improved PCL mechanical properties; however, higher doses would decrease them. The amorphous character of the PCL decreases and crystallinity increases with an increase in dose (15; 25; 35 kGy), which may be due to the scissioning of the polymer chains, through which the polymer undergoes some spatial rearrangement and the small fragments may rearrange themselves towards a new crystalline zone. The higher dose (65 kGy), on the other hand, resulted in the decrease in crystallinity due to the crosslinking of fragmented chains, which changed the regularly arranged crystallites into non-arranged ones by forming new bonds between the neighboring chains [25]. The authors observed that the tensile strength increased as the irradiation dose increased. However, the maximum elongation showed no significant variation as the radiation dose increased. In general, both chain scission and crosslinking take place simultaneously. It is plausible to say that the chain scission process is yet to take prominence at low doses (15, 25, 35 kGy). The fragmented chains undergo crystallization to a larger extent than the crosslinked and large macromolecular assemblies. PCL membranes are effectively sterilized by irradiating with 35 kGy of gamma exposure, which is a suitable dose that does not compromise the materials’ properties or cell proliferation [25]. Other authors [25] claim that the effect of irradiating PCL fibers, irrespective of dosage, caused a significant reduction in both the fibers’ stiffness and strength. The maximum elongation (strain) was achieved by the different fiber groups resulted in no significant differences when compared with the unirradiated scaffolds. De Cassan et al. report that some mechanical properties were not completely in line with the findings of other authors. This was probably due to inconsistent electrospinning and tensile testing protocols [22]. Thus, results for the effect of gamma radiation on the mechanical properties are ambiguous and a general trend has not yet been established.

Nonetheless, considering the exploratory nature of this work, further assays are required to properly evaluate the properties of the material exposed to different gamma-radiation intensities. Techniques such as gel permeation chromatography (GPC), solid-state nuclear magnetic resonance (NMR) spectroscopy, differential scanning calorimetry (DSC) and degradation tests will complement the results obtained at this stage and will allow a more in-depth and unequivocal characterization of the explored biomaterials in terms of their physicochemical characteristics and degradation pattern after exposure to gamma radiation. Thus, the authors intend to continue this study by applying these production and sterilization conditions to customized 3D-printed PCL scaffolds for trachea occlusion.

## Figures and Tables

**Figure 1 ijms-23-15916-f001:**
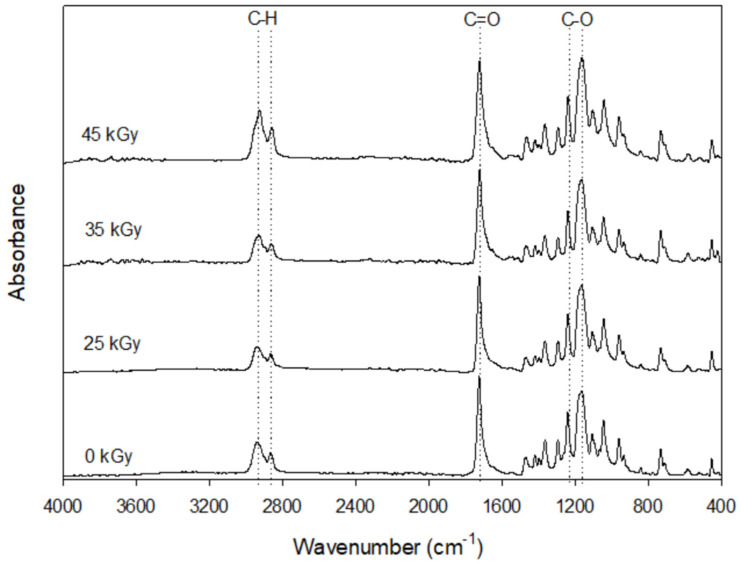
FTIR spectra of PCL scaffolds sterilized with different gamma radiation conditions.

**Figure 2 ijms-23-15916-f002:**
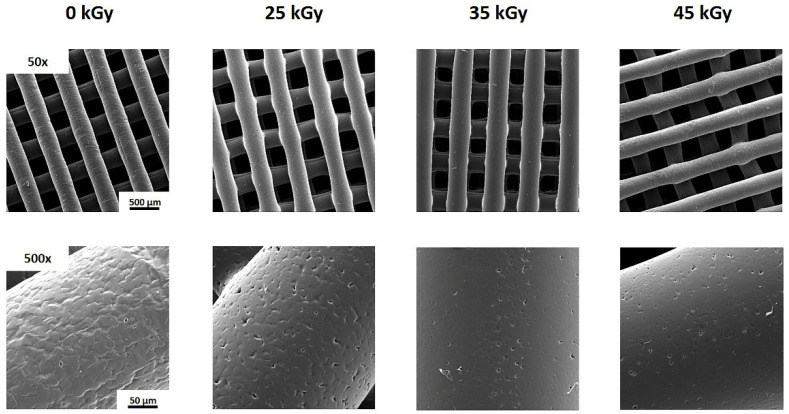
SEM images of gamma sterilized and unsterilized PCL scaffolds.

**Figure 3 ijms-23-15916-f003:**
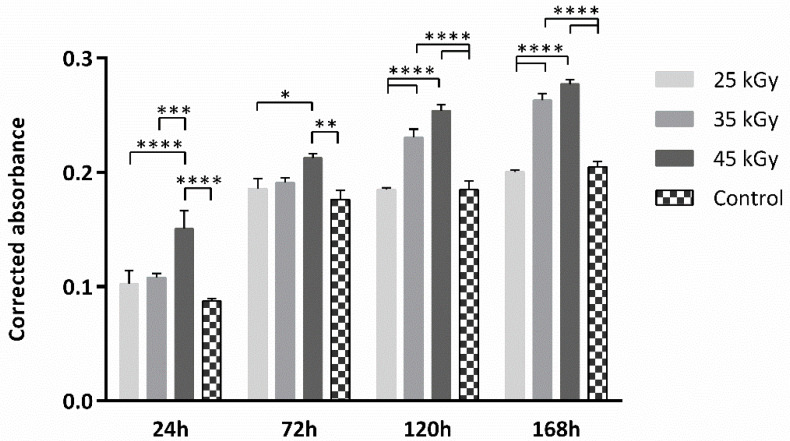
Presto Blue^TM^ cytocompatibility assessment with hDPSCs. Results presented as mean ± SE. Results’ significance is presented through the symbol (*), according to the *p* value, with one, two, three or four symbols, corresponding to 0.01 < *p* ≤ 0.05; 0.001 < *p* ≤ 0.01; 0.0001 < *p* ≤ 0.001; and *p* ≤ 0.0001, respectively.

**Figure 4 ijms-23-15916-f004:**
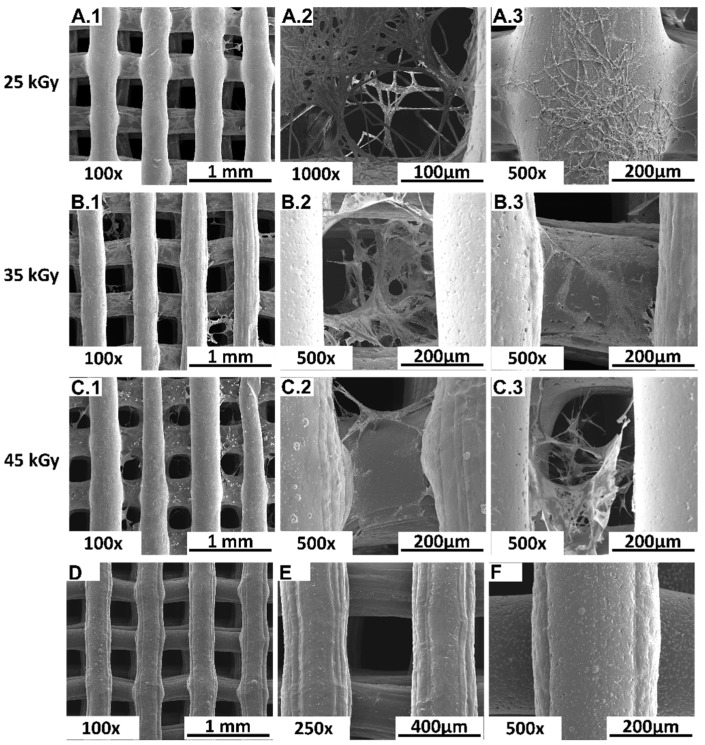
SEM images obtained from the 3D PCL devices seeded with hDPSCs. (**A1**–**C3**) represent seeded scaffolds sterilized with 25 kGy gamma radiation 35 kGy and 45 kGy, respectively. (**D**–**F**) represent the unseeded devices, sterilized with 25 kGy gamma radiation 35 kGy and 45 kGy, respectively. Left panel with 100× magnification and middle and right panel with 500× magnification.

**Figure 5 ijms-23-15916-f005:**
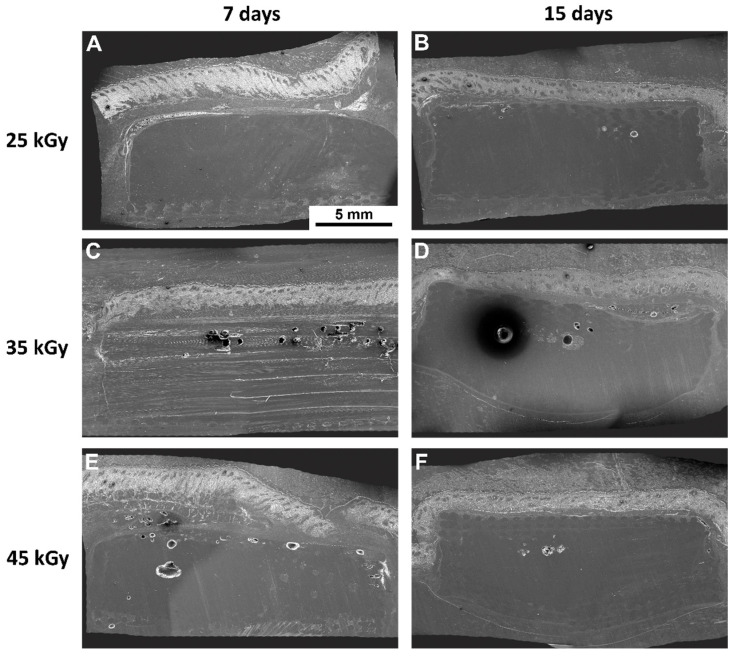
SEM images of the subcutaneously implanted 3D PCL devices in the rat animal model. Representing the left panel and the right panel, 7 days, and 15 days recovery time, respectively. (**A**,**B**) devices were sterilized with 25 kGy gamma radiation, (**C**,**D**) with 35 kGy and (**E**,**F**) with 45 kGy. Magnification of 20× and scale bar 5 mm.

**Figure 6 ijms-23-15916-f006:**
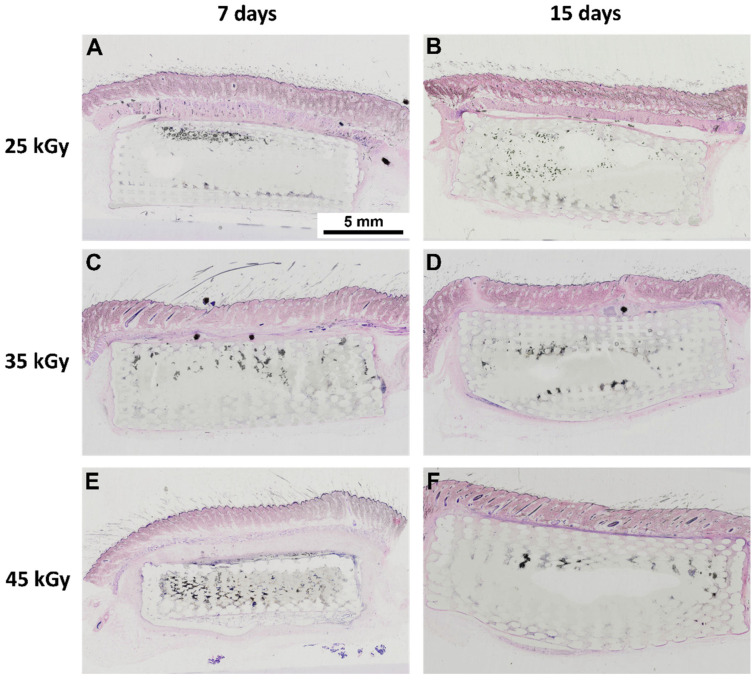
Histological images, stained with H&E, scanned using an Olympus Virtual Microscopy System VS110TM at 20× magnification, of the subcutaneously implanted 3D PCL scaffolds in the rat animal model. Representing the left panel and the right panel, 7 days, and 15 days recovery time, respectively. (**A**,**B**) scaffolds were sterilized with 25 kGy gamma radiation, (**C**,**D**) with 35 kGy and (**E**,**F**) with 45 kGy. Scale bar 5 mm.

**Figure 7 ijms-23-15916-f007:**
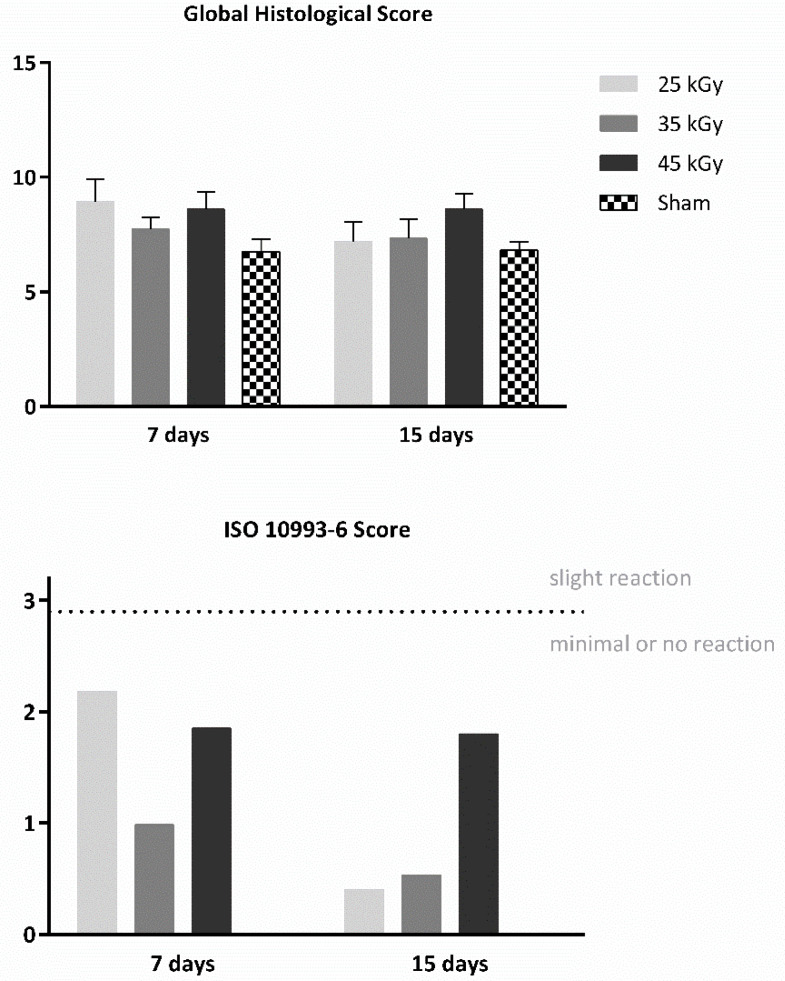
Global histological score (upper panel) and calculated score, according to guideline ISO 10993-6:2016 (lower panel), for subcutaneous implantation (biocompatibility assessment) of the PCL devices, 7 and 15 days after implantation time.

**Table 1 ijms-23-15916-t001:** Presto Blue^TM^ cytocompatibility assessment with hDPSCs. Results presented as mean ± SE.

	25 kGy	35 kGy	45 kGy	Control
24 h	0.102 ± 0.012	0.108 ± 0.003	0.151 ± 0.016	0.087 ± 0.002
72 h	0.186 ± 0.009	0.191 ± 0.004	0.213 ± 0.004	0.176 ± 0.008
120 h	0.185 ± 0.002	0.231 ± 0.007	0.254 ± 0.005	0.185 ± 0.007
168 h	0.200 ± 0.002	0.263 ± 0.006	0.277 ± 0.004	0.205 ± 0.005

**Table 2 ijms-23-15916-t002:** Global histological scores presented as mean ± SE following ISO-10993-6 guidelines for the PCL devices groups and sham, at 7 and 15 days after implantation.

	Sham	25 kGy	35 kGy	45 kGy
7 days	6.750 ± 0.532	8.933 ± 0.959	7.733 ± 0.521	8.600 ± 0.748
ISO SCORE	-	2.183	0.983	1.850
15 days	6.800 ± 0.396	7.200 ± 0.846	7.333 ± 0.826	8.600 ± 0.675
ISO SCORE	-	0.400	0.533	1.800

## Data Availability

The data that support the findings of this study are available from the corresponding author on request.

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
