# Peer review of "Gamma Irradiation Processing on 3D PCL Devices—A Preliminary Biocompatibility Assessment"

_ijms, 2022, doi:10.3390/ijms232415916_

Round 1

Reviewer 1 Report

Thank you for your high-level results of the research. Your study is that the surface of PCL is modified with increasing radiation dose, then PCL was improved biocompatibility. 

However, Several researchers have been studied the effect of gamma irradiation on different organic and inorganic loaded polymeric materials. a number of literature have announced that increasing irradiation will promote or reduce the physicomechanical performance of PCL. 

Therefore, I recommended that you will perform additional research or recommend that show an optimal combination that increases biocompatibility and prevents deterioration of physicomechanical properties among the differences level of sterilization

Author Response

Answer to reviewer 1

Dear reviewer 1:

Thank you very much for the feedback on this review phase, and also for the suggestions made, which received the best attention from us. The authors inform that the final document has been revised and all suggestions made by the reviewers have been introduced, being duly identified in the final document with highlight in different colors. This review also made it possible to identify and correct some errors and typos as well as improve the general level of English.

The changes made to the document are described below. All changes introduced and highlighted text segments appear in the final document highlighted in blue.

  • However, several researchers have been studied the effect of gamma irradiation on different organic and inorganic loaded polymeric materials. a number of literature have announced that increasing irradiation will promote or reduce the physicomechanical performance of PCL.

Therefore, I recommended that you will perform additional research or recommend that show an optimal combination that increases biocompatibility and prevents deterioration of physicomechanical properties among the differences level of sterilization

Answer :

The authors understand the suggestions raised by the Reviewer.

The authors recognize that additional studies are necessary to assess the physical-mechanical properties and deterioration of the material after exposure to different intensities of gamma radiation. For this deeper assessment, techniques such as gel permeation chromatography (GPC), Solid-state Nuclear Magnetic Resonance spectroscopy (NMR), Differential Scanning Calorimetry (DSC) and degradation tests would be an asset in the present work. Although these are valuable techniques that would improve the evaluation of scaffolds, as mentioned in the manuscript, this is considered a preliminary study that aims to evaluate the cytocompatibility and biocompatibility of polycaprolactone devices after different gamma radiation conditions. Nevertheless, as a further work, we will perform a more in-depth analysis of the materials, going through the techniques mentioned above.

Furthermore, considering the reviewer's suggestions, the following has been added to the conclusions:

“Considering mechanical properties, yield stress increased significatively but not the stress at break. In the case of the scaffold it represents an important support rule as in the airway, this is of the upmost relevance [24]. Augustine et al report a low radiation dose first would lead to improved PCL mechanical properties, however, higher doses would decrease them. The amorphous character of the PCL decreases and crystallinity increases with increase in dose (15; 25; 35kGy), which may be due to the scissioning of the polymer chains, by which the polymer undergoes some spatial rearrangement. The small fragments may rearrange themselves towards new crystalline zone. Whereas the higher dose (65 kGy) resulted in the decrease in crystallinity due to the crosslinking of fragmented chains, which change the regularly arranged crystallites into non-arranged ones by forming new bonds between the neighboring chains [20]. They observed that as the irradiation dose increases the tensile strength increased. However, the maximum elongation showed no significant variation as the radiation dose increases. In general, both chain scission and crosslinking take place simultaneously. It is plausible to say that the chain scission process is yet to take prominence at low doses (15, 25, 35kGy). The fragmented chains undergo crystallization in a large extent than the cross-linked and large macromolecular assemblies. PCL membranes are effectively sterilized by irradiating with 35kGy of gamma exposure, which is a suitable dose without compromising the materials properties as well as cell proliferation [20]. Other authors [21] describe that the effect of irradiating PCL fibers, irrespective of dosage, caused a significant reduction in both the fibers’ stiffness and strength. The maximum elongation (strain) was achieved by the different fiber groups resulted in no significant differences when compared with the unirradiated scaffolds. De Cassan et al report that some mechanical properties were not completely in line with the findings of other authors. This was probably due to inconsistent electrospinning and tensile testing protocols [17]. Thus, results for the effect of gamma radiation on the mechanical properties are ambiguous and a general trend has not yet been established.”

“Nonetheless, further research is required to evaluate the properties of the material exposed to different radiation intensities. Techniques such as gel permeation chromatography (GPC), solid-state nuclear magnetic resonance (NMR) spectroscopy, differential scanning calorimetry (DSC) and degradation tests will complement the current study in the future.”

Regarding the reviewer's question concerning the increase of biocompatibility and avoid the deterioration of the physicomechanical properties of the material, the combination of PCL with hydroxyapatite would be an advantage.

According to the literature, the hydroxyapatite, presents good stability, biocompatibility and degradability, it also promotes the cell adhesion/proliferation. Sousa and colleagues demonstrated that the addition of hydroxyapatite to PCL decreases the contact angle, consequently, it can increase cellular activity, namely cell adhesion and proliferation, and, therefore, potentially make a composite more appropriate in the tissue engineering field [1].

[1] Sousa, A.C.; Biscaia, S.; Alvites, R.; Branquinho, M.; Lopes, B.; Sousa, P.; Valente, J.; Franco, M.; Santos, J.D.; Mendonça, C.; Atayde, L.; Alves, N.; Maurício, A.C. Assessment of 3D-Printed Polycaprolactone, Hydroxyapatite Nanoparticles and Diacrylate Poly(ethylene glycol) Scaffolds for Bone Regeneration. Pharmaceutics 2022, 14, 2643. https://doi.org/10.3390/pharmaceutics14122643

Reviewer 2 Report

In the article, the authors investigate and try to define the ideal conditions for the sterilization of 3D polycaprolactone implants using radiation. This topic is very current and essential for further applied research.

The submitted manuscript is qualitatively very successful, but it contains some shortcomings that would be good to solve before the actual publication.

Title and Keywords: I see the password 3D polycaprolactone as problematic, I would rather replace it with 3D printing, and polycaprolactone.

In the introduction, it would be good to mention some other important properties of polycaprolactone (molecular weight and its connection with the degradation profile, etc..)

The authors refer to the sample production methodology using the Biomate technology described in reference 25, however, this reference refers to the Biomate technology to another reference in Appl. Moss. Mater. 2019. This is not appropriate, the authors should use the correct reference or better describe the fabrication of the samples with this technology at least in the supplementary. Overall, it would be appropriate to zoom in on the owl and the entire technological concept.

The authors state in the introduction that the conclusions of the effect of radiation on material and mechanical properties are ambiguous. Then it is very important within this article to verify some of the effects of radiation on PCL in order to be able to declare conclusions about the correct sterilization procedure:

 The FTIR method is not accurate enough to detect these effects. It would be good to determine the potential molecular weight change using GPC. Potential degradation or alteration of PCL molecules, for example, using solid state NMR.

Another suitable method could be the use of DSC to reveal the potential change in PCL crystallinity.

These tested properties have a direct impact on the degradation profile of PCL as a biodegradable polymer, and this must be known for further application steps.

The authors report that the samples showed better cell viability when using 35 and 45 kGy radiation compared to 25 kGy. Are the authors able to discuss the mechanism why this is so? Which factors have changed at higher radiation levels and are involved in increasing cell viability.

Author Response

Answer to reviewer 2

Dear reviewer 2:

Thank you very much for the feedback on this review phase, and also for the suggestions made, which received the best attention from us. The authors inform that the final document has been revised and all suggestions made by the reviewers have been introduced, being duly identified in the final document with highlight in different colors. This review also made it possible to identify and correct some errors and typos as well as improve the general level of English.

The changes made to the document are described below. All changes introduced and highlighted text segments appear in the final document highlighted in blue.

  • Title and Keywords: I see the password 3D polycaprolactone as problematic, I would rather replace it with 3D printing, and polycaprolactone.

Answer:

 The suggestion was accepted and the key word “3D polycaprolactone” replaced by “3D printing; polycaprolactone”.

  • In the introduction, it would be good to mention some other important properties of polycaprolactone (molecular weight and its connection with the degradation profile, etc..)

Answer:

The authors agree with the reviewer's suggestion. The following text has been introduced into the document:

“Polycaprolactone (PCL) is a biodegradable polyester with a molecular wight of 114.142 g mL-1,a density of 1.146 g/mL at 25 °C, a low melting point of around 60 °C and a glass transition temperature of about −60 °C. It can be used as a polymeric plasticizer because of its ability to lower elastic modulus and soften other polymers. Its surface is chemistry suitable for cell attachment, proliferation, and differentiation, and its degradation by-products are nontoxic and are usually metabolized and eliminated via natural pathways.

Woodward and his group studied the in vivo and intracellular degradation of PCL and reported that degradation first occur with nonenzymatic bulk hydrolysis; a transient initial inflammatory response occurred only for the first 2 weeks. After 9 months, when the molecular weight had significatively reduce, a loss in mass emerge and PCL did fragment[18].

The most important properties of a bioabsorbable scaffold are the degradation rate, mechanical strength, and ability to support cell growth. Gamma rays at 30.8kGy significantly decrease the rate of degradation of PCL, without affecting molecular weight nor cells attachment and growth. Other studies of various doses of gamma irradiation and the impact on PCL revealed a shift toward a lower molecular weight in a dose-dependent manner accompanied by an increase in both the melting point and crystallinity [18]. In addition, D'alelio et al report typical critical doses (200-300kGy (20-30Mrad)) required to initiate gel formation in linear polyesters containing 3-7 methylene groups per ester group [22]. Narkis et al report g.p.c molecular weight distributions as a function of the irradiation dose in the pregelation stage of PCL (Dose < 260kGy (26Mrad)). Number and weight average molecular having CH2/COO ratio of infinity undergoes only crosslinking reactions, and gel formation is found from the very beginning of the irradiation process if the starting molecular weight is not too low. Typical critical dose of 260kGy (26 Mrad) is required to initiate gel formation in PCL (CH2/COO = 5/1) [23]. Despite of gamma rays have induced chain scission and crosslinking, these works are focussed mainly on much higher doses than that required for sterilisation.”

  1. Woodward, S.C., Brewer, P.S., Moatamed, F., Schindler, A. and Pitt, C.G. The intracellular degradation of poly(ε-caprolactone). J. Biomed. Mater. 1985; Res., 19: 437-444. https://doi.org/10.1002/jbm.820190408.
  • The authors refer to the sample production methodology using the Biomate technology described in reference 25, however, this reference refers to the Biomate technology to another reference in Appl. Moss. Mater. 2019. This is not appropriate, the authors should use the correct reference or better describe the fabrication of the samples with this technology at least in the supplementary. Overall, it would be appropriate to zoom in on the owl and the entire technological concept.

Answer:

The authors agree with the reviewer's suggestion and propose the following:

In the current research, a biomanufacturing system (ANI Biomate Project) developed by CDRSP-IPLeiria was used. This system incorporates several techniques such as: micro -extrusion, multi-head dispensing and electrospinning [1]. The extrusion technique used in this study consisted of heating the material (PCL), and pneumatic and mechanical extrusion, i.e., compressed air and a rotating spindle were used for controlled deposition of the composite, respectively.”

[36] Biscaia, S.; Dabrowska, E.; Tojeira, A.; Horta, J.; Carreira, P.; Morouço, P.; Mateus, A.; Alves, N. Development of heterogeneous structures with Polycaprolactone-Alginate using a new 3D printing system–BioMEDβeta: Design and processing. Procedia Manuf. 2017, 12, 113–119.

Reference 25 was replaced by the proper citation: [25] Viana T, Biscaia S, Dabrowska E, Franco MC, Carreira P, Morouço P, and Alves, N. A Novel Biomanufacturing System to Produce Multi-Material Scaffolds for Tissue Engineering: Concept and Preliminary Results. Applied Mechanics and Materials; 2019: Trans Tech Publ. https://doi.org/10.4028/www.scientific.net/AMM.890.283”

  • The authors state in the introduction that the conclusions of the effect of radiation on material and mechanical properties are ambiguous. Then it is very important within this article to verify some of the effects of radiation on PCL in order to be able to declare conclusions about the correct sterilization procedure:

 The FTIR method is not accurate enough to detect these effects. It would be good to determine the potential molecular weight change using GPC. Potential degradation or alteration of PCL molecules, for example, using solid state NMR. Another suitable method could be the use of DSC to reveal the potential change in PCL crystallinity. These tested properties have a direct impact on the degradation profile of PCL as a biodegradable polymer, and this must be known for further application steps.

Answer:

The authors agree with the suggestions made by the Reviewer.

The authors recognize that further studies are required to completely evaluate and characterize the physical-mechanical properties and the degradation pattern of the material after exposure to different intensities of gamma-radiation. For this deeper assessment, techniques such as gel permeation chromatography (GPC), Solid-state Nuclear Magnetic Resonance spectroscopy (NMR), Differential scanning calorimetry (DSC) and degradation tests would be an asset in the present work.

The GPC technique is a valuable analysis that allows the complete characterization of the molecular weight distribution of a polymeric material. 

The solid-state NMR spectroscopy is a technique that allows, at the atomic level, to determine the chemical structure, 3D structure and dynamics of PCL scaffolds.

The DSC is a powerful method of thermal analysis in which the heat flux entering/leaving the material is measured as a function of temperature or time, while the sample is exposed to a temperature program. This technique evaluates various material properties, including glass transition temperature, melting, crystallization, specific heat capacity, oxidation behavior, and thermal stability.

Although these are valuable techniques that would improve the evaluation of scaffolds, this work is a preliminary study that aims to evaluate the cytocompatibility and biocompatibility of polycaprolactone devices after exposure to different gamma radiation conditions. Nevertheless, in later stages of this project, more in-depth analysis of the materials is planned, in which the techniques suggested by the reviewer and referred to above will certainly be explored.

Furthermore, considering the reviewer's suggestions, the following text segment has been added to final document in the in the “conclusions” section:

The suggestion was accepted and the following text was also included in the manuscript:

“Considering mechanical properties, yield stress increased significatively but not the stress at break. In the case of the scaffold it represents an important support rule as in the airway, this is of the upmost relevance [24]. Augustine et al report a low radiation dose first would lead to improved PCL mechanical properties, however, higher doses would decrease them. The amorphous character of the PCL decreases and crystallinity increases with increase in dose (15; 25; 35kGy), which may be due to the scissioning of the polymer chains, by which the polymer undergoes some spatial rearrangement. The small fragments may rearrange themselves towards new crystalline zone. Whereas the higher dose (65 kGy) resulted in the decrease in crystallinity due to the crosslinking of fragmented chains, which change the regularly arranged crystallites into non-arranged ones by forming new bonds between the neighboring chains [20]. They observed that as the irradiation dose increases the tensile strength increased. However, the maximum elongation showed no significant variation as the radiation dose increases. In general, both chain scission and crosslinking take place simultaneously. It is plausible to say that the chain scission process is yet to take prominence at low doses (15, 25, 35kGy). The fragmented chains undergo crystallization in a large extent than the cross-linked and large macromolecular assemblies. PCL membranes are effectively sterilized by irradiating with 35kGy of gamma exposure, which is a suitable dose without compromising the materials properties as well as cell proliferation [20]. Other authors [21] describe that the effect of irradiating PCL fibers, irrespective of dosage, caused a significant reduction in both the fibers’ stiffness and strength. The maximum elongation (strain) was achieved by the different fiber groups resulted in no significant differences when compared with the unirradiated scaffolds. De Cassan et al report that some mechanical properties were not completely in line with the findings of other authors. This was probably due to inconsistent electrospinning and tensile testing protocols [17]. Thus, results for the effect of gamma radiation on the mechanical properties are ambiguous and a general trend has not yet been established.”

Nonetheless, considering the exploratory nature of this work, further assays are required to properly evaluate the properties of the material exposed to different gamma-radiation intensities. Techniques such as gel permeation chromatography (GPC), solid-state nuclear magnetic resonance (NMR) spectroscopy, differential scanning calorimetry (DSC) and degradation tests will complement the results obtained at this stage and will allow a more in-depth and unequivocal characterization of the explored biomaterials in terms of their physical-chemical characteristics and degradation pattern after exposure to gamma-radiation.”

  • The authors report that the samples showed better cell viability when using 35 and 45 kGy radiation compared to 25 kGy. Are the authors able to discuss the mechanism why this is so? Which factors have changed at higher radiation levels and are involved in increasing cell viability.

Answer:

The following text segment has been introduced into the document:

“These results have already been identified in previous studies with the same tendency, that is, with materials subjected to higher radiation levels showing a higher rate of cell proliferation and cytocompatibility [31, 32]. These results confirm that gamma radiation has no adverse effects on cell proliferation and promotes cell proliferation in a dose-dependent manner. Although the reason for this observation is unclear, previous work seems to indicate that an increase in radiation dose is associated with changes in the hydrophilicity of materials [31]. Higher doses of radiation seem to promote a decrease in the contact angle, which in turn stimulates adhesion and consequent cell proliferation. However, a more extensive characterization of the devices would be necessary to confirm this biophysical change and fully justify the results.”

Round 2

Reviewer 2 Report

Thanks to the authors for taking the comments into account.

Based on the changes, I recommend the manuscript for publication.

Before that, however, I recommend a small correction of the part about the description of PCL in the introduction:

PCL does not have only one specific mean  value of MW (weight no wight) as the authors state. PCL is normally synthesized with molecular weight values of 10,000 - 90,000, this value has a direct effect on the degradation profile in the range from approx. 1 - 4 years. This needs to be put into perspective. Along with this, typical crystallinity values should also be mentioned.

Author Response

Answer to reviewer 2

Dear reviewer 2:

Thank you very much for the feedback on this review phase, and also for the suggestions made, which received the best attention from us. The authors inform that the final document has been revised and all suggestions made by the reviewers have been introduced, being duly identified in the final document with highlight in different colors. This review also made it possible to identify and correct some errors and typos as well as improve the general level of English.

The changes made to the document are described below. All changes introduced and highlighted text segments appear in the final document highlighted in blue.

  • Based on the changes, I recommend the manuscript for publication. Before that, however, I recommend a small correction of the part about the description of PCL in the introduction: PCL does not have only one specific mean value of MW (weight no wight) as the author’s state. PCL is normally synthesized with molecular weight values of 10,000 - 90,000, this value has a direct effect on the degradation profile in the range from approx. 1 - 4 years. This needs to be put into perspective. Along with this, typical crystallinity values should also be mentioned.

Answer:

The authors agree with the reviewer's suggestion. The following text has been introduced into the document:

“Polycaprolactone (PCL) is a hydrophobic, biodegradable polyester with a molecular wight usually between 3000 and 80000 g/mol, a density of 1.146 g/mL at 25 °C, a low melting point of around 60 °C and a glass transition temperature of about −60 °C. It is a semi-crystalline polymer in which its crystallinity tends to decrease with increasing molecular weight [18-20]. A study by Castilla-Cortázar et al. calculated a percentage crystallinity in pure PCL of 39.1 by differential scanning calorimetry analysis [21]. The total degradation of PCL is considerably affected by the molecular weight and crystallinity of the material and can vary between one and four years. It can be used as a polymeric plasticizer because of its ability to lower elastic modulus and soften other polymers. Its surface is chemistry suitable for cell attachment, proliferation, and differentiation, and its degradation by-products are nontoxic and are usually metabolized and eliminated via natural pathways.

Woodward and his group studied the in vivo and intracellular degradation of PCL and reported that degradation first occur with nonenzymatic bulk hydrolysis; a transient initial inflammatory response occurred only for the first 2 weeks. After 9 months, when the molecular weight had significatively reduce, a loss in mass emerge and PCL did fragment [22].”

[18] Baptista, C.; Azagury, A.; Shin, H.; Baker, C.M.; Ly, E.; Lee, R.; Mathiowitz, E. The effect of temperature and pressure on polycaprolactone morphology. Polymer 2020, 191, 122227, doi:https://doi.org/10.1016/j.polymer.2020.122227.

[19] Woodruff, M.A.; Hutmacher, D.W. The return of a forgotten polymer—Polycaprolactone in the 21st century. Progress in Polymer Science 2010, 35, 1217-1256, doi:https://doi.org/10.1016/j.progpolymsci.2010.04.002.

[20] Contardi M, Alfaro-Pulido A, Picone P, Guzman-Puyol S, Goldoni L, Benítez JJ, Heredia A, Barthel MJ, Ceseracciu L, Cusimano G, Brancato OR, Di Carlo M, Athanassiou A, Heredia-Guerrero JA. Low molecular weight ε-caprolactone-p-coumaric acid copolymers as potential biomaterials for skin regeneration applications. PLoS One. 2019 Apr 8;14(4):e0214956. doi: 10.1371/journal.pone.0214956. PMID: 30958838; PMCID: PMC6453441.

[21] Castilla-Cortázar, I.; Vidaurre, A.; Marí, B.; Campillo-Fernández, A.J. Morphology, Crystallinity, and Molecular Weight of Poly(ε-caprolactone)/Graphene Oxide Hybrids. Polymers (Basel) 2019, 11, doi:10.3390/polym11071099.

[22] Woodward, S.C., Brewer, P.S., Moatamed, F., Schindler, A. and Pitt, C.G. The intracellular degradation of poly(ε-caprolactone). J. Biomed. Mater. 1985; Res., 19: 437-444. https://doi.org/10.1002/jbm.820190408
